Yu *et al. Genome Biology*    (2020) 21:94

SOFTWARE

# scATAC-pro: a comprehensive workbench for single-cell chromatin accessibility sequencing data

Wenbao Yu[1,2], Yasin Uzun[1,2], Qin Zhu[3], Changya Chen[1,2] and Kai Tan[1,2,3,4*]

* Correspondence: tank1@email.chop.edu
[1]Center for Childhood Cancer Research, The Children's Hospital of Philadelphia, Philadelphia, PA 19104, USA
[2]Department of Biomedical and Health Informatics, The Children's Hospital of Philadelphia, Philadelphia, PA 19104, USA
Full list of author information is available at the end of the article

## Abstract

Single-cell chromatin accessibility sequencing has become a powerful technology for understanding epigenetic heterogeneity of complex tissues. However, there is a lack of open-source software for comprehensive processing, analysis, and visualization of such data generated using all existing experimental protocols. Here, we present scATAC-pro for quality assessment, analysis, and visualization of single-cell chromatin accessibility sequencing data. scATAC-pro computes a range of quality control metrics for several key steps of experimental protocols, with a flexible choice of methods. It generates summary reports for both quality assessment and downstream analysis. scATAC-pro is available at https://github.com/tanlabcode/scATAC-pro.

**Keywords:** Chromatin accessibility, Single cell, Genomics, Bioinformatics

## Background

Chromatin accessibility is a strong indicator of the activities of functional DNA sequences. Recently, multiple experimental protocols have been developed to profile genome-wide chromatin accessibility in single cells, including the Assay of Transposase Accessible Chromatin with high-throughput sequencing (scATAC-seq) [1], single-cell combinatorial indexing ATAC-seq (sci-ATAC-seq) [2], single-cell transposome hypersensitivity site sequencing (scTHS-seq) [3], plate-based scATAC-seq protocol [4], and droplet-based single-cell combinatorial indexing ATAC-seq (dsci-ATAC-seq) [5]. In this paper, we collectively define data generated with these experiment protocols as single-cell chromatin accessibility sequencing data, or scCAS data. Application of these protocols has helped to understand the epigenetic heterogeneity across cell populations in complex tissues during normal development and pathogenesis, including adult mouse tissues [6], forebrain development [7], hematopoietic differentiation and leukemia evolution [8, 9], and T cell development and exhaustion [10].

In contrast to the rapid growth of scCAS data, bioinformatics tools for scCAS data analysis are critically lacking. The majority of existing tools lack comprehensiveness in their ability to process scCAS data. Both chromVar [11] and single-cell regulome

analysis toolbox (SCRAT) [12] work with preprocessed data and only report loss or gain of chromatin accessibility on a set of predefined genomic regions, which ignores a large amount of information in the data. Detection of cell-type-specific difference in chromatin accessibility (Detin) [13], single-cell accessibility-based clustering (scABC) [14], and cisTopic [15] focuses on identifying cell populations and/or differential accessible regions given the processed data such as bam files or in peak-by-cell count matrix.

To our knowledge, Scasat [16] and scitools [17] are the only published software for comprehensive analysis of scCAS data. However, Scasat is developed in the Jupyter notebook environment. Although it is interactive, the programming codes are hard to standardize and reuse and users need to customize the analysis step by step. Furthermore, Scasat binarizes raw peak-by-cell count matrix, which ignores the differences among accessible regions and thus may lead to loss of valuable information for downstream analysis. Scasat does not provide summary reports for either data quality assessment or downstream analysis. scitools is designed for sci-ATAC-seq data only. Another tool SnapATAC [18] also binarizes the raw count data and cluster cells based on bin-by-cell count matrix. Cellranger-atac by 10x Genomics is another comprehensive tool but only works with data generated using the 10x Genomics platform, and the software code is not open source. Additionally, some key analysis modules of cellranger-atac are not flexible and do not use state-of-the-art algorithms. For example, the peak calling module does not use state-of-the-art algorithms, such as model-based analysis for ChIP-Seq 2 (MACS2) [19], resulting in many problematic peaks.

Here, we present a comprehensive and open-source software package for quality assessment and analysis of single-cell chromatin accessibility data, scATAC-pro. It provides flexible options for most of the analysis modules with carefully curated default settings. Summary reports for both quality assessment and downstream analysis are automatically generated. Interface to an interactive single-cell data exploration and visualization tool VisCello [20] is also provided.

## Results

### Overview of scATAC-pro workflow

scATAC-pro consists of two units, the data processing unit and the downstream analysis unit (Fig. 1). The data processing unit takes raw fastq files for reads and barcodes as the input and outputs peak-by-cell count matrix, QC report, and genome track files. It consists of the following modules: demultiplexing, adaptor trimming, read mapping, peak calling, cell calling, genome track file generation, and quality control assessment. The downstream analysis unit consists of the following modules: dimension reduction, cell clustering, differential accessibility analysis, Gene Ontology analysis, TF motif enrichment analysis, TF footprinting analysis, prediction of chromatin interactions, and integration of multiple data sets. We provide flexible options for all modules. Details about each module are summarized in Additional file 2: Table S1. We designed the scATAC-pro to be user friendly. In each run, users just need to specify the input file ("--*input*" flag), the module name ("--*step*" flag), and a configuration file ("--*config*" flag) in which users provide parameters and options for the analysis modules. Users can choose to run the entire or partial workflow. By default, all results are saved in the "output" directory under the current directory (--*output_dir* "./output").

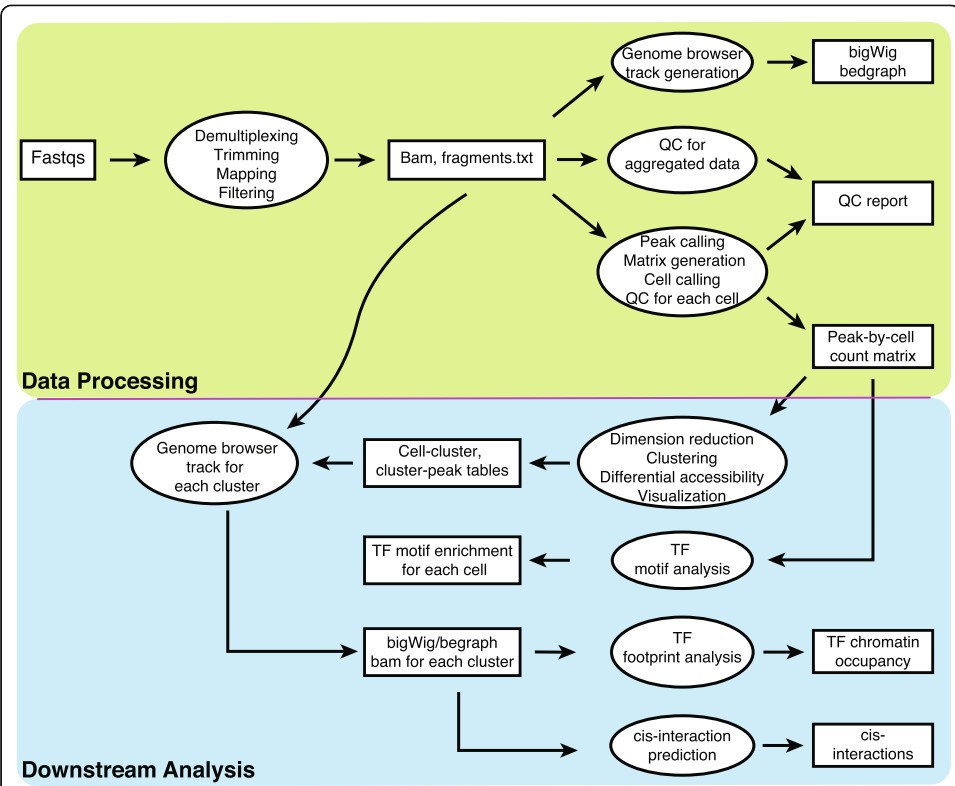

**Fig. 1** The scATAC-pro workflow. The workbench consists of two units, data processing unit and downstream analysis unit. Modules of the data processing unit include demultiplexing and adaptor trimming of the raw reads, followed by mapping of reads to the reference genome and filtering of low-quality reads. Aggregated reads are used for generating genome browser tracks, peak calling, and cell calling. Quality check (QC) reports are generated based on both aggregated data and single-cell data. Modules of the downstream analysis unit consists of dimension reduction, clustering, differential accessibility between different cell populations, genome browser track generation per cell subpopulation, TF motif enrichment analysis, footprinting analysis, and prediction of chromatin interactions. Most modules provide more than one analysis methods

## scATAC-pro provides flexible choices of methods for many analysis tasks

We provide at least two methods for most data processing and analysis modules. There are several reasons to have multiple methods for a given data processing or analysis task. First, for certain tasks such as read mapping, many tools such as BWA [21], Bowtie [22], and Bowtie2 [23] exist that perform equally well but have different levels of trade-off between mapping accuracy and mapping speed [23, 24]. Users can choose among those aligners based on different goals. For cell calling, cells can be called by filtering low-quality barcodes [1, 2, 6, 9] or by using model-based approaches (e.g., cellranger-atac). Each category of methods has its pros and cons that can be tailored towards the goals of the analysis.

Second, in cases where a general-purpose and top-performing method exists, such as MACS2 [19] for peak calling, there are other methods that are more suitable for specific tasks. For example, the genome-wide event finding and motif discovery algorithm, GEM [25], was shown to have better performance in identifying peaks overlapping with transcription factor binding sites (TFBS) [26]. In this case, the users might prefer GEM over MACS2.

Third, often times, there is a need for methods that can address data set-specific characteristics. For example, using both real and simulated data, we found that the clustering

performance using binarized/non-binarized data varies among different methods. For instance, cisTopic performs better using binarized data whereas SCRAT performs better using non-binarized data (Additional file 1: Fig. S1A, Fig. S2D). Even for the same clustering algorithm, we found that most of the time, both Louvain (Additional file 1: Fig. S1B) and K-mean (Additional file 1: Fig. S1C) clustering algorithms work better with non-binarized data although at higher noise levels, the performance difference becomes smaller, and in some cases using binarized data is more accurate. Given this observation, we provide methods that work on either binarized or non-binarized data or both (see "Methods" for details). Another example is clustering, a critical task for understanding heterogeneity in a cell population. To help select better clustering methods, we conducted benchmarking studies using simulated data and real data. The compared methods include scABC, chromVAR, cisTopic, latent semantic indexing or LSI [2], SCRAT, and Louvain algorithm implemented in Seurat v3 [27]. Based on the benchmarking result (Additional file 1: Fig. S2), we recommend cisTopic, SCRAT, and Seurat as the top-performance methods although all above methods are available in scATAC-pro.

### scATAC-pro provides carefully evaluated default settings for all modules

Method and parameter choice makes a big difference in the result of several analysis modules. We therefore provide a set of carefully evaluated default settings for each analysis module. We discuss the settings for the major modules as follows (see "Methods" for details).

#### Read mapping

Because of its balance between mapping speed and accuracy [28], especially for paired-end sequencing data, we choose BWA (specifically bwa-mem) as the default read aligner.

#### Peak calling

Peak calling is usually done on aggregated data across all barcodes. Such an approach fails to identify peaks that only appear in rare cell populations. We implemented a two-step strategy, similar to the idea used by Cusanovich et al. [6]. We first segment the genome into 5-kb bins and generate a bin-by-barcode count matrix, removing barcodes with fewer than 1000 unique fragments. We then cluster the barcodes using the graph-based Louvain algorithm using principal components as the input. Finally, we use MACS2 to call peaks on the aggregated data for each cell cluster. The final set of peaks are generated by merging peaks less than 200 bp apart identified from different cell clusters.

#### Cell calling

As default, we use a filtering strategy to distinguish cell barcodes from non-cell barcodes, because the method is intuitive, easy to interpret, and widely used among published studies [1, 2, 4, 6, 9, 17]. We define a barcode as a cell if its total number of unique fragments is greater than 5000 and the fraction of such fragments in peaks is greater than 50%. Users can use different thresholds for the fraction of fragments in enhancers, promoters, or mitochondrial genome to filter barcodes as well.

### Normalization

We provide two normalization methods. The term frequency-inverse document frequency (TF-IDF) method [2, 6] treats count data as binary and normalizes data by sequencing depth per cell and total number of unique fragments per peak. In the second method, count data is first log-transformed, followed by a linear regression to remove the confounding factor due to varying sequencing depth per cell for every peak. This method enables users to work with non-binarized count data. The TF-IDF method is set as the default normalization method in scATAC-pro.

### Dimension reduction and data visualization

We use principal component analysis (PCA) as the default dimension reduction method because it is the most widely used method for scCAS data and easy to interpret. Note that if the PCA is conducted on the TF-IDF normalized data, such dimension reduction is also referred as latent semantic indexing or LSI [2]. We use the PCA implementation in Seurat v3 with some modifications. In Seurat v3, the raw count matrix is log-transformed followed by a regression to remove confounding factors (the total number of unique fragments per cell). PCA is then performed on the transformed features. Because there are usually hundreds of thousand peaks in a scCAS data set, this process takes about a couple of hours to finish for a typical data set. In our implementation, we first perform PCA on the normalized peak-by-cell count matrix, followed by a regression analysis on each principal component. This procedure substantially reduces the computation time and produces very similar clustering results as the original Seurat implementation (Additional file 1: Fig. S3). Uniform Manifold Approximation and Projection (UMAP) [29] is used as the default visualization method.

### Clustering

We provide the graph-based Louvain algorithm implemented in Seurat v3 as the default clustering method. Shared neighbor network (SNN) graph is constructed based on the first 30 principal components. Louvain algorithm is then performed on the SNN graph with default setting. We found that the Louvain clustering algorithm has a better balance between accuracy and running speed among several popular clustering methods (Additional file 1: Fig. S2).

## scATAC-pro provides summary reports and interface to visualization tools

scATAC-pro generates quality assessment metrics for both aggregated and single-cell data. Two types of metrics are generated. The first type of metrics evaluates data quality internally, including read mapping rate, duplicate rate, high-confidence mapped fragments (MAPQ score greater than 30), and library complexity. The second type of metrics evaluates data quality using external annotations of genomic features, including fraction of fragments in mitochondrial genome and fraction of fragments overlapping with peaks and other annotated genomic regions, such as enhancers and promoters. The quality assessment summary reports are generated in html format. These statistics can be used to filter low-quality barcodes.

Besides quality assessment metrics, scATAC-pro also generates summary reports for downstream analyses, including dimension reduction, clustering, differential chromatin

accessibility analysis, TF motif enrichment analysis and footprinting analysis, gene ontology analysis, and prediction of chromatin interactions.

To enable interactive exploration of data in scATAC-pro, we provide an interface to VisCello [20], a data explorer and visualization tool for single-cell omics data. To do this, we annotate the peaks with its nearest gene and mark genes with their TSSs located within the peak. Users can then visualize the chromatin accessibility signal of each peak or gene in all cells and identify differential accessible peaks among arbitrary groups of cells.

## scATAC-pro provides utility functions to facilitate downstream analyses

### Generation of input files for genome browser tracks for each cluster

In addition to visualizing scCAS signal in VisCello, it is also common to visualize scCAS signal for each cell population on a genome browser. To generate a normalized signal track file, bam file of cells from each cell cluster is first split from the bam file of all barcodes. Reads per cluster are then normalized as reads per kilobase per million mapped reads. scATAC-pro outputs normalized chromatin accessibility for each cluster in bigWig and bedGraph file formats, which can be directly uploaded to a genome browser for visualization.

### Transcription factor footprinting analysis

ATAC-seq and related technologies use the Tn5 enzyme to cleavage nucleosome-free DNA while keeping the transcription factor binding sites intact due to protection by the bound TF. As a result, a small region, referred to as the footprint, exhibits reduced Tn5 cleavage rate at the ATAC-seq peak locus. Unlike TF motif enrichment analysis, TF footprinting analysis provides direct evidence of TF binding to the chromatin [30]. With Hint-ATAC [31], scATAC-pro enables footprinting analysis of either one cell cluster or differential TF binding between two groups of cell clusters.

### Integration of multiple scCAS data sets

To integrate multiple scCAS data sets, assuming each data set is processed by scATAC-pro, we first merge peaks identified in each data set. Using this merged set of peaks, scATAC-pro reconstructs the peak-by-cell matrix for each data set. Because we generate the peak-by-cell matrix using the same set of peaks for all samples, it is straightforward to integrate the data sets using existing tools, such as Seurat v3 or Harmony [32]. scATAC-pro uses Seurat v3 as the default for this task.

### Peak annotation and gene ontology analysis

To facilitate Gene Ontology analysis of genes associated with differential accessibility peaks, scATAC-pro first annotates each peak with its nearest gene. Gene Ontology analysis for those genes can then be performed using the *runGO* module. The background gene set is comprised of all genes associated with the differential accessibility peaks in all cell groups resulted from the differential accessibility analysis. This analysis helps users to further explore the identity of each cell cluster.

### Predicting chromatin interactions by Cicero

Connecting regulatory DNA elements to target genes is a prerequisite to understanding transcriptional regulation. Cicero [33] predicts the interactions between cis-regulatory elements and the target genes using scCAS data. scATAC-pro generates the predicted interactions by running the *runCicero* module. The resulting interactions can be viewed through the UCSC genome browser.

### Case study

We used three real data sets generated with different experimental protocols to demonstrate the utility of scATAC-pro: a data set of sorted human bone marrow hematopoietic cells generated using the Fluidigm protocol [8, 9] (Buenrostro2018), a data set of human peripheral blood mononuclear cells (PBMCs) generated using the 10x Genomics protocol [34], and a data set of 13 adult mouse tissues generated using the sci-ATAC-seq protocol [6] (Cusanovich2018). For the sake of brevity, we present the result based on the 10x Genomics data set in the main figures. Similar figures based on the two other data sets are presented as supplementary figures.

Starting from the fastq files, scATAC-pro first demultiplexed sequencing reads by adding the cell barcodes (R2.fastq.gz) information to the paired-end reads (R1.fastq.gz, R3.fastq.gz). Adaptor sequences were then trimmed off, mapped to the GRCh38 reference genome using scATAC-pro default settings. Fragments with mapping quality score (MAPQ score) less than 30 were removed. A summary report for mapping statistics and library complexity is provided for all reads (Fig. 2A) and reads belonging to called cells (Fig. 2B, C). Summary reports for the other two data sets are shown in Additional file 1: Fig. S4 and Fig. S5, respectively.

Using the default peak caller, scATAC-pro called 129,049 peaks after removing peaks overlapping with ENCODE blacklisted genomic regions [35]. Cell barcodes were selected by filtering out barcodes with fewer than 5000 total unique fragments and the fraction of unique fragments in peak less than 50% (Fig. 3A). Quality assessment report for each barcode was generated using various metrics, including distribution of insert size, transcriptional start site (TSS) enrichment profile, distribution of the total number of unique fragments for cell and non-cell barcodes, and fractions of unique fragments overlapping with annotated genomic regions (Fig. 3B–E). Overall statistics of data aggregated from all called cells was also computed (Fig. 3F). Figures showing quality assessment metrics for called single cells for the other two data sets are shown in Additional file 1: Fig. S6 and Fig. S7, respectively.

Downstream analyses including clustering, TF motif enrichment analysis, TF footprinting analysis, GO analysis, and cis-element interaction prediction were conducted using default scATAC-pro methods and settings (Fig. 4). In total, we found 10 cell types. The top 10 enriched TFs for each cluster are shown in Fig. 4B, which provides a means for identifying cell type associated with each cluster. For example, binding motifs of PU.1 (encoded by *SPI1*), IRF4, CEBPA, and CEBPB are highly enriched in clusters 0, 6, 7, and 8, suggesting those clusters are monocytes [36]. Motifs of EOMES and TBX5 were enriched in clusters 1, 2, and 5, suggesting those clusters are T cells. Enrichment of EBF1 [37] and BCL11A motifs [38] suggests cluster 3 represents B cells. The differential footprinting analysis between cluster 0 and the rest of clusters further

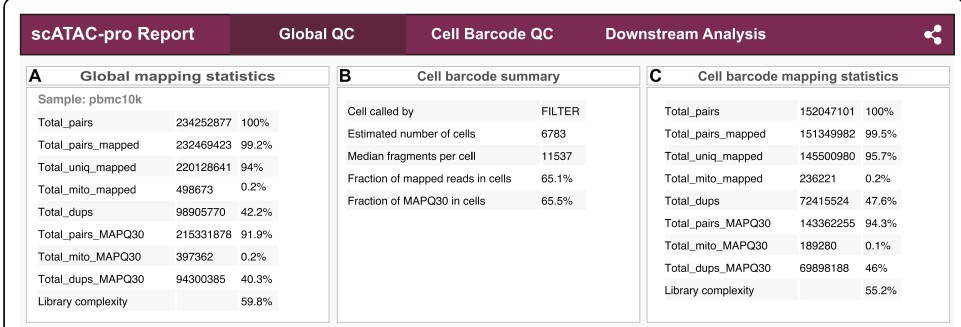

**Fig. 2** Summary statistics for read mapping, library complexity, and cell calling. scATAC-seq data of human peripheral blood mononuclear cells (PBMCs) was used for illustration purpose. Global mapping statistics are based on all data (A). Cell barcode mapping statistics are based on called cells (B, C). MAPQ, mapping quality score

suggests that cluster 0 represents monocytes, because the monocytic TFs PU.1, JUNB, JUN, CEBPA, and CEBPE [36, 39, 40] all have significant higher binding probability in cluster 0 cells (Fig. 4C).

Summary report for downstream analyses for the other two data sets is shown in Additional file 1: Fig. S8 and Fig. S9, respectively.

Using VisCello [20], we can display chromatin accessibility values of TSS regions of several marker genes across cell clusters (Fig. 5A and Additional file 1: Fig. S10A), such as *MS4A1* (CD20) for B cells, *GNLY* and *NKG7* for natural killer (NK) cells, *CD3E* for T cells, *CD14*, *LYZ*, and *FCGR3A* (CD16) for monocyte cells, and *CST3* for dendritic cells (DC) [41]. We also displayed UCSC genome browser tracks for two example genes, *CD14* (Fig. 5B) and the *FCER1A* (Additional file 1: Fig. S11). Taken together, based on the chromatin accessibility profile of known cell-type-specific marker genes,

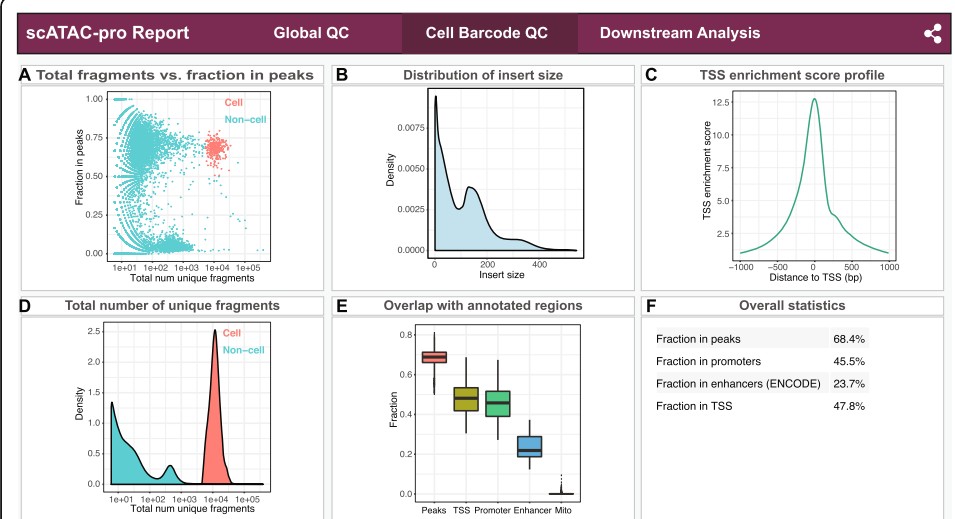

**Fig. 3** Quality assessment metrics for called single cells. scATAC-seq data of human PBMCs was used for illustration purpose. (A) Plot of the fraction of fragments in peaks versus the total number of unique fragments. The plot can be used to distinguish cell barcodes from non-cell barcodes. (B) Distribution of insert fragment sizes. The plot can be used to evaluate the quality of transposase reaction. (C) Transcription start site (TSS) enrichment profile. (D) Distribution of the total number of unique fragments for cell and non-cell barcodes. The plot can be used to evaluate the amount of cell debris sequenced. (E) Boxplot of fragments overlapping annotated genomic regions per cell. (F) Overall statistics of data aggregated from all called cells

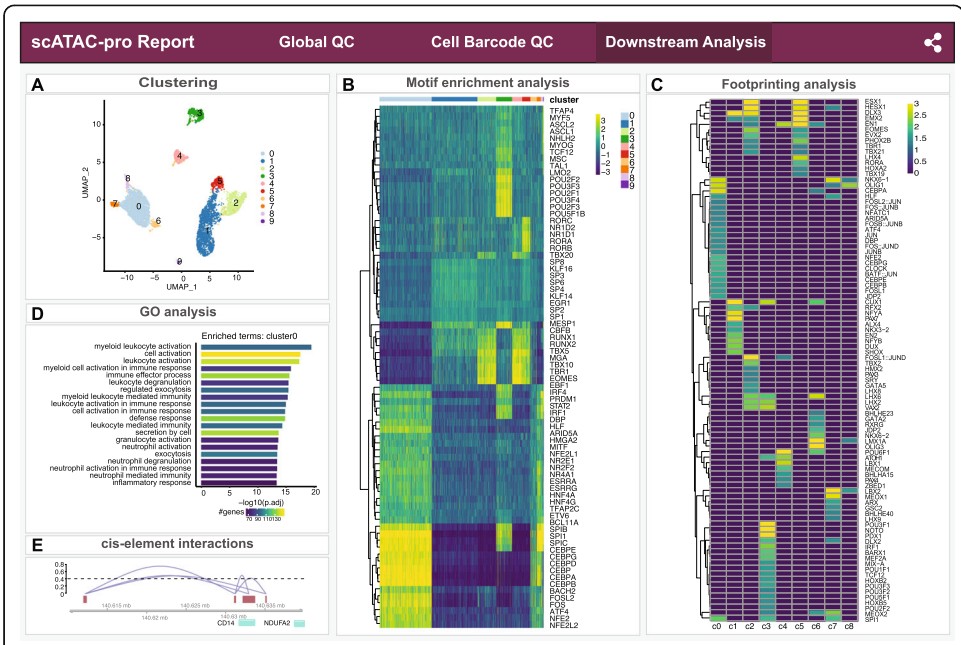

**Fig. 4** Summary report for downstream analyses of human PBMC data set. Results of the following analyses are shown: clustering analysis (A), transcription factor (TF) motif enrichment analysis (B), differential footprinting analysis (between one cluster and the rest of the clusters (C), enriched gene ontology (GO) terms for cluster0 (D), and predicted cis-interactions at the *CD14* locus (E)

we annotated cell clusters as T cells, B cells, CD14$^+$ monocytes, CD16$^+$ monocytes, dendritic cells, and natural killer cells (Fig. 5C). VisCello can also conduct differential chromatin accessibility analysis between any two arbitrary groups of cell clusters (Additional file 1: Fig. S10B).

## Discussion

scATAC-pro provides a comprehensive solution for scCAS data QC and analysis. It reports a number of QC metrics for both aggregated data of all barcodes and barcodes of called cells. These metrics evaluate multiple steps of the experimental protocol, including the quality of nuclei preparation, transposase reaction, and cell encapsulation (insert size, fraction of unique reads, cell vs non-cell reads, mitochondrial reads), and library preparation and sequencing (duplicate rate, fraction of unique reads and reads with MAP score > 30). Although there is no universally optimal QC metric for all types of scCAS data, the fraction of fragments in peak per cell is the most widely used in the literature [1, 2, 4, 6, 9, 17]. Alternative metric such as the TSS enrichment score per cell is introduced recently [18], but its utility may be limited for cell types that have a large fraction of active TSS-distal peaks. Having a comprehensive annotation of cis-regulatory elements across all human cell types will facilitate the task of evaluating quality of gene-distal ATAC-seq peaks.

Because there is no clear optimal method for many analysis tasks, scATAC-pro provides multiple methods that allow users to tailor their analyses and to address data set-specific characteristics. To guide the users, we have provided carefully evaluated default settings for each analysis task, including both method choice and parameter setting of the selected method(s).

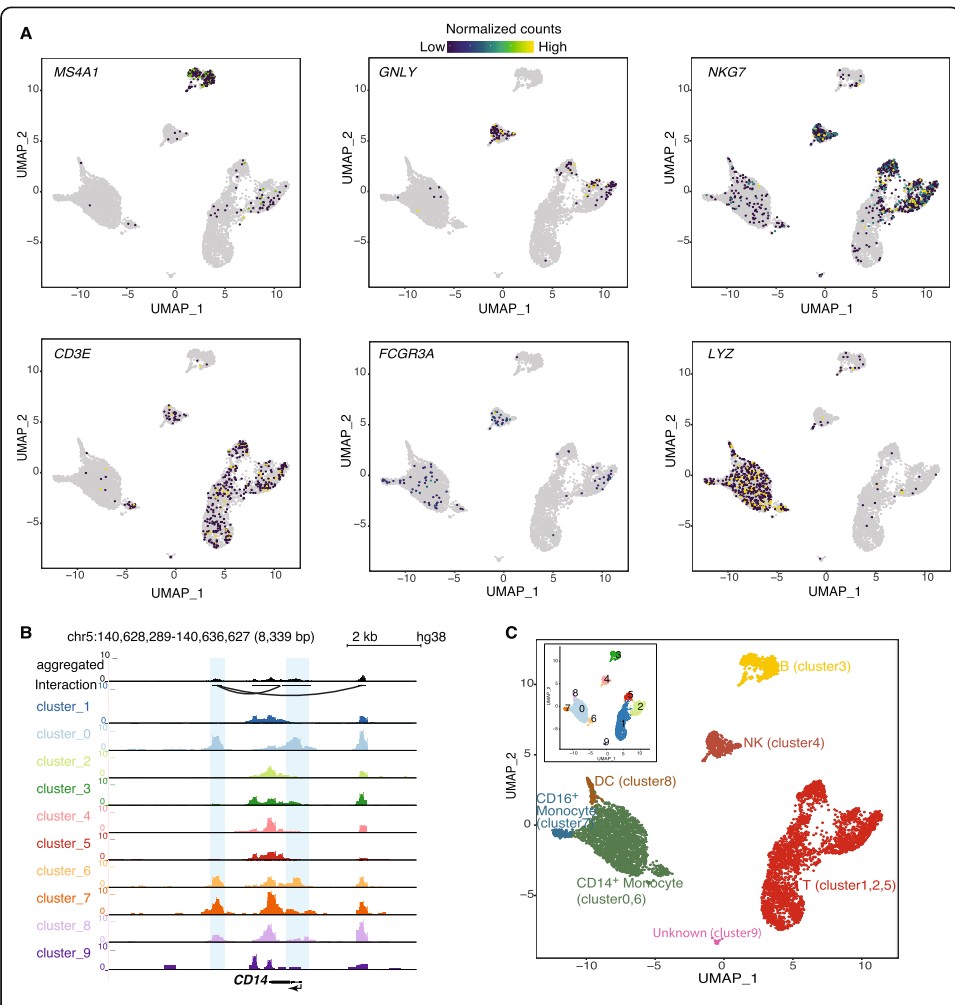

**Fig. 5** Visualization of scATAC-seq data. (A) Chromatin accessibility signal of single cells. Normalized chromatin accessibility signal for peaks overlapping with transcriptional start sites of selected marker genes. Data is visualized by automatically calling VisCello. (B) Chromatin accessibility signal of aggregated cells. Genome browser view of normalized chromatin accessibility signal at the *CD14* locus across cell clusters. (C) Cell type assignment based on chromatin accessibility signals of known cell type marker genes. Inset, clustering result without cell type assignment

The open-source and modular design of scATAC-pro facilitates the maintenance and future development of the software. Several experimental protocols exist for generating scCAS data. Data generated using these protocols have different characteristics and qualities. scATAC-pro is the first software tool that enables analysis of all types of scCAS data. By doing so, scATAC-pro facilitates the integration of rapidly growing scCAS data.

The current version of scATAC-pro generates a static summary report. It can be enhanced by generating dynamic summary report in future versions of the software. For example, for downstream analysis, the results can be updated in real time based on the cell clusters compared. Interactively display of cis-interactions within an arbitrary genomic region will be another very useful feature to add.

## Conclusions

scATAC-pro is a comprehensive open-source software for processing, analyzing, and visualizing single-cell chromatin accessibility sequence data. With the rapid accumulation

of single-cell chromatin accessibility sequencing data, application of scATAC-pro will facilitate a better understanding of epigenomic heterogeneity in healthy and diseased tissues.

## Methods

### Details of each scATAC-pro module

#### Demultiplexing

Given the reads fastq files and the barcode fastq files, the barcode sequences are written into the name of each read sequence (in the format of @BARCODE:ORIGINAL_READ_ NAME) to facilitate the tasks of downstream modules, such as generating peak-by-cell matrix and quality assessment of single cells. For data generated using 10x Genomics, sci-ATAC-seq, and dsci-ATAC-seq protocols, users need to provide the paired-end read fastq files and the index fastq file (also supports multiple index fastq files). For scTHS-seq data, users need to specify the parameter *isSingleEnd = TRUE* in the configuration file because scTHS-seq data are single-end reads. This module is skipped if the barcode for each read is recorded in the required format. For example, in the mouse sci-ATAC-seq atlas data [2, 6], the barcode for each read is saved in the name of each read.

#### Adaptor trimming

To map sequencing reads confidently to the reference genome, scATAC-pro first trims off the adaptor sequence and primer oligo sequence from raw reads using trim_galore *(http://www.bioinformatics.babraham.ac.uk/projects/trim_galore/)* as the default, which can automatically detect and trim the adaptor and primer sequences. Alternatively, users can also use Trimmomatic [42], which is faster but users need to specify the sequence of the adaptor in the configuration file.

#### Read mapping

Different alignment methods make different compromises between mapping accuracy and speed. BWA, Bowtie, and Bowtie2 are three popular and top-ranked aligners based on previous benchmarking studies [24, 28]. scATAC-pro enables all three aligners for read mapping. We use BWA (bwa-mem) as the default aligner based on its balance between mapping speed and accuracy. Users can provide additional options in the configuration file by specifying *MAPPTING_METHOD* and corresponding parameters. For instance, if users want to use 10 CPUs for parallel computing, they can set *BWA_OPTS = -t 10* if BWA is used, *BOWTIE_OPTS = -p 10* if BOWTIE is used, and *BOWTIE2_OPTS = -p 10* if Bowtie2 is used. After mapping, scATAC-pro uses samtools [43] to sort, index, mark duplicates. and filter low-quality reads in the bam file.

The position sorted bam file, filtered bam file (default MAPQ score > 30), and the mapping statistics are automatically generated and saved for downstream modules. A file called *fragment.txt* that records the genomic location, barcode, and the number of duplicates of each unique fragment is generated using a custom R script to facilitate downstream analysis.

#### Peak calling

By default, scATAC-pro calls peaks using aggregated fragments across all barcodes. MACS2 is a popular peak calling tool for ATAC-seq and ChIP-Seq data. We also

enable the GEM algorithm for peak calling. It is recommended by the ENCODE consortium for its good performance on calling peaks with enriched TF motifs. The processed scCAS data is then summarized as the peak-by-barcode matrix. Peaks that only appear in rare cell types are challenging to call by the above approach of using aggregated fragments across all barcodes. An alternative approach is to bin the genome without peak calling or combination of binning the genome and peak calling [6]. For the combination strategy, we first segment the genome into 5-kb bins and generate bin-by-barcode count matrix, removing barcodes with low total number of fragments (e.g., 1000). We then cluster the barcodes followed by peak calling for each cluster using MACS2. Peaks or bins overlapped with blacklisted genomic regions are removed for downstream analysis. Users can specify *PEAK_CALLER* to be one of MACS2, GEM, BIN, or COMBINED in the configuration file.

### Cell calling

Not all barcodes are real cells in a typical scCAS data set due to cell collision and/or cell debris. How to distinguish cell barcodes from non-cell barcodes is still a challenging problem. Generally, users select cell barcodes either by filtering out low-quality barcodes based on some summary statistics, such as total number of fragments and fraction of fragments in peak regions. Alternatively, users can use model-based approaches. For example, the cellRanger-atac method fits a mixture of two zero-inflated negative binomial models to discriminate between cell barcodes and non-cell barcodes. EmptyDrops [44], originally designed to identify cells from scRNA-Seq data, models the counts using a Dirichlet-multinomial distribution. scATAC-pro provides all of the aforementioned strategies/methods. Based on our experience, cellranger-atac and EmptyDrops (with the default fdr of 0.001) tend to call too many cells, while the knee point approach of EmptyDrops and cellRanger-atac are too stringent. Therefore, we choose the filtering strategy as the default since it is simple and intuitive. For the filtering strategy, users can filter barcodes based on one or multiple summary statistics such as the total number of unique fragments, fraction of fragments in peaks, fraction of fragments in mitochondrial genome, and fraction of fragments overlapping with annotated promoters, enhancers, and TSS regions. Since the implementation of cellRanger-atac cell calling is not publicly available, we implemented the algorithm using custom R scripts.

### Quality assessment

scATAC-pro provides mapping statistics for all reads as well as reads belonging to called cells. The following QC metrics are reported: total reads, total number of mapped reads, unique mapping rate, fraction of reads in mitochondrial genome, number of duplicate reads, high-quality reads (MAPQ > 30), library complexity, fraction of reads in annotated genomic regions, and TSS enrichment profile. The same set of summary statistics is also reported for reads belonging to called cells. scATAC-pro also reports the number of cells called, median number of fragments per cell, and fraction of mapped reads belonging to cells.

### Normalization

The default term frequency-inverse document frequency (TF-IDF) normalization is implemented using the TF-IDF function in Seurat v3. We also provide an alternative

normalization method, which first log-transforms the count followed by regression to remove the confounding factor due to sequencing depth per cell.

### Dimension reduction, cell clustering, and visualization

scATAC-pro supports principal component analysis (PCA) (which is also called latent semantic indexing (LSI) if the data were first normalized by TF-IDF) and latent Dirichlet allocation (LDA) for dimension reduction. We use the Seurat v3 toolkit to implement PCA, Louvain clustering algorithm, and the cisTopic R package to implement LDA. Different numbers of reduced dimensions can be specified by *nREDUCTION* parameter (default 30) in the configuration file. We provide t-distributed stochastic neighbor embedding (tSNE) and uniform manifold approximation and projection (UMAP) (implemented in Seurat v3) for visualization.

### Differential chromatin accessibility analysis

Peaks with differential accessibility across different cell clusters are potentially cell-specific gene regulatory elements. We use Wilcoxon test as the default method to perform differential accessibility analysis. Alternative methods such as logistic regression-based method (implemented in Seurat v3), DESeq2 [45], and negative binomial regression-based test (implemented in Seurat v3) are also available. Users can compare two clusters or one cluster vs the rest of clusters using the module *runDA* and specify *group1* and *group2* in the configuration file.

### Generation of genome browser track files

scATAC-pro outputs bigWig and bedGraph files for visualizing chromatin accessibility signal in a genome browser. The signal is normalized by reads per kilobase per million mapped reads. Those files are generated using the bamCoverage command in deepTools toolkit [46].

### TF motif enrichment analysis

scATAC-pro constructs the TF binding accessibility profile for each single cell using chromVAR with a slight modification. chromVAR computes a gain or loss of accessibility score for peaks sharing the same motif by comparing accessibility scores of peaks with similar mean accessibility and GC contents. To speed up this analysis, in scATAC-pro, instead of using the whole peak-by-cell matrix, we select the top 30% of most variable peaks. This reduces the running time of chromVAR by eight times compared to using the full matrix of the PBMC data. We then identify TFs that have significantly higher accessibility in one cell cluster than in the other cell cluster by conducting a two-sample Wilcoxon test. TFs that have significantly higher chromatin accessible in each cell cluster are saved in a text file and visualized using a heatmap.

### TF footprinting analysis

We use Hint-ATAC [31] to perform TF footprinting analysis, which is the first tool designed specifically for ATAC-seq data. Due to the sparsity of scCAS data, it is impossible to predict TF footprints at a single-cell level, but feasible at cell cluster level since the read depth per cluster is comparable to bulk ATAC-seq data. We also use Hint-

ATAC to do differential TF footprinting analysis, by which users can find differentially bound TFs between any two cell clusters or between one cell clusters and the rest of the cell clusters.

### Summary reports

scATAC-pro automatically updates the summary report after data processing, and downstream analyses were done using custom R scripts. If some analysis modules are not executed, scATAC-pro still generates the report with results of the executed modules.

### Clustering methods used in published tools for scCAS data

#### Louvain algorithm implemented in Seurat v3

The Louvain algorithm for community detection is a popular algorithm for detecting communities in a network. It maximizes a modularity score for each community. We used the FindCluster function in Seurat to implement this algorithm, which takes data in reduced dimensions as the input (the first 30 PCs as default). The function requires a resolution parameter to control the number of clusters indirectly, where a larger resolution parameter results in a larger number of clusters.

#### cisTopic

cisTopic takes the binary peak-by-cell count matrix as the input and conducts latent Dirichlet allocation (LDA) analysis to produce topic-by-cell and peak-by-topic probability matrices. The LDA is also a method of dimensionality reduction; therefore, different numbers of topics can be used. We run cisTopic with 10, 20, 30, 50, 80, and 100 topics and then use the seleModel function to select the best number of topics. Hierarchical clustering (hclust function in R) is then performed on the topic-by-cell probability matrix.

#### chromVAR

chromVAR computes a gain or loss of accessibility score for peaks sharing the same TF motif by comparing accessibility scores of peaks with similar mean accessibility and GC contents. A TF-by-cell $z$-score matrix is then calculated, indicating the enrichment of each TF at peaks for each single cell. We conduct PCA on the TF-by-cell $z$-score matrix. Hierarchical clustering is performed on the first 20 PCs.

#### scABC

scABC weighs cells by the number of distinct reads within $+/-$ 500-kb peak regions and then applies a weighted K-medoids clustering to partition the cells into clusters.

#### LSI

LSI first filters out peaks that are accessible in fewer than 150 cells and then normalizes the filtered peak-by-cell matrix using the TF-IDF algorithm. The singular value decomposition (SVD) is performed, and the hierarchical clustering is run on the 2nd to 10th PCs.

### SCRAT

SCRAT constructs features as either peaks or regions of interest (e.g., gene sets or genomic regions overlapping with TF motifs). Dimension reduction is done on PCA or tSNE, and multiple conventional clustering methods, including mclust, hierarchical clustering, and K-means clustering, can be applied to the reduced dimensions. Here, we implemented the default setting, by which the raw peak-by-cell matrix is normalized by library size and then PCA is conducted, the final clustering is done on the first 20 PCs using the mclust algorithm.

### Simulated data with different levels of noise using bulk ATAC-seq data

We generated simulated data with different levels of noise from the bulk ATAC-seq data of 13 primary human blood cell types [8] using the same strategy as that in [47]. We started with the bulk peak-by-cell count matrix and generated count for peak $i$ in cell type $t$ using a binomial distribution $binom(2, p_i^t)$, where $p_i^t = (1-q)r_i^t/2 + qn/2k$, $r_i^t$ is the percentage of all reads overlapping with peak $i$ in cell type $t$, $k$ is the total number of peaks in the bulk data, $n$ is the number of simulated fragments, and $q$ is a parameter specifying the level of noise; $q = 0$ indicates no noise while $q = 1$ indicates the highest level of noise.

### Simulated data by subsampling from bulk ATAC-seq data

The single-cell data were simulated by resampling bulk ATAC-seq data described above. Specifically, we simulated data for 200 cells for each of the 13 cell types. For each cell, 10,000 reads were randomly selected from the mapped reads in the bulk data. Peaks were called using MACS2 using the aggregated single-cell data. The performance of each method was evaluated using the adjusted rand index and bulk sorted cell types as the ground truth (Additional file 1: Fig. S2). To investigate the robustness of each method in the presence of varying cell type compositions, we sampled a total of 1000 cells from the 13 cell types with different cell type compositions. The fractions of different cell types were generated based on the Dirichlet distribution (with shape parameter alpha = 3 for each component). For each clustering method, the default parameters were used, except for the number of clusters, which was set to 13.

### Supplementary information

---

**Additional file 1:** Figs. S1–11.

**Additional file 2:** Table S1.

**Additional file 3:** Review History.

---

#### Acknowledgements
We thank the Research Information Services at the Children's Hospital of Philadelphia for providing computing support.

#### Peer review information

#### Review history
The review history is available as Additional file 3.

#### Funding
This work was supported by National Institutes of Health of United States of America grants HD089245, CA226187, and CA233285 (to K.T.), a grant from the Leona M. and Harry B Helmsley Charitable Trust (2008-04062 to K.T.), and a grant from the Alex's Lemonade Stand Foundation (to K.T.).

### Availability of data and materials

The scATAC-pro software is freely available under the MIT license. Source code has been deposited at the GitHub repository (https://github.com/tanlabcode/scATAC-pro) [48] and Zenodo with the access code DOI: https://doi.org/10.5281/zenodo.3696036 [49]. To improve the portability of the software, we also provide a docker version of the software at https://hub.docker.com/r/wbaopaul/scatac-pro.

Accession numbers for third party data used in this study include GSE74912, GSE111586, and GSE96769. Additional simulated data sets and analysis code are described in this published article and deposited at the GitHub repository (https://github.com/tanlabcode/scATAC-pro_paper) [50] and Zenodo with the access code DOI: https://doi.org/10.5281/zenodo.3732194 [51].

### Authors' contributions

WY and KT conceived and designed the study. WY designed and implemented the scATAC-pro software with the help of YU, QZ, and CC. YU and QZ provided additional analytical tools. WY performed data analysis. KT supervised the overall study. WY and KT wrote the paper. The author(s) read and approved the final manuscript.

### Ethics approval and consent to participate

Not applicable.

### Consent for publication

Not applicable.

### Competing interests

The authors declare that they have no competing interests.

### Author details

<sup>1</sup>Center for Childhood Cancer Research, The Children's Hospital of Philadelphia, Philadelphia, PA 19104, USA.
<sup>2</sup>Department of Biomedical and Health Informatics, The Children's Hospital of Philadelphia, Philadelphia, PA 19104, USA. <sup>3</sup>Genomics and Computational Biology Graduate Group, University of Pennsylvania, Philadelphia, PA, USA.
<sup>4</sup>Department of Pediatrics, University of Pennsylvania, Philadelphia, PA 19104, USA.

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

## 