## [**Additional file 3:** Review History. · Genome Biology]

Review History

First round of review

Reviewer 1

Were you able to assess all statistics in the manuscript, including the appropriateness of statistical tests used?

There are no statistics in the manuscript.

Were you able to directly test the software?

Yes

Comments to author:

In this manuscript, Yu et al. presented a software package for scATAC-seq. Single cell epigenome sequencing techniques like scATAC-seq is the cutting edge methods to investigate the functional network beneath the gene expression regulation. However, emerging scATAC-seq platforms and tools may imposed a heavy learning burden on the non-specialists when analyzing scATAC-seq data. Thus, an integrated workbench for scATAC-seq is welcomed. The software package scATAC-pro was designed in comprehensive function, covering nearly the full pipeline of scATAC-seq analysis, from raw read pre-processing to downstream functional/TF analysis. The manuscript is well organized and clearly written. Nevertheless, there are several prominent technical problems to be settled.

Major points:

- 1) I do not agree with the authors' argument that data binarization of scATAC-seq is not a good choice (as it misses important information). Binarization is helpful to reduce noise and bias in the sparsely mapped scATAC-seq data. The authors would consider to use additional simulated data with noise and/or with more biased mapping to genome regions to evaluate the robustness of their selected methods.
- 2) In addition to simulated data, at least one real-world dataset should be used for the method evaluation. Such benchmark settings could be found in a recently published work: Chen et al. Assessment of computational methods for the analysis of single-cell ATAC-seq data. *Genome Biol* 24: 241.
- 3) It is not clear how the authors optimized the parameters or tool selection for different scATAC-seq platforms. I have only noticed the specification of single-end/paired-end reads for this purpose. How about the fitting to different read length / mapping rate / depth etc. between various platforms?
- 4) Data visualization functions was not readily accessible to the potential users of scATAC-pro.

To this end, automatically calling of genome browser and Viscello tool is preferred, but a step-by-step guideline to configure and run these tools based on scATAC-pro results should be the minimum requirement.

5) Batch effect has been noted in scATAC-seq (e.g. Baker et al., Classifying cells with Scasat, a single-cell ATAC-seq analysis tool. Nucl Acids Res, 2019). It is not clear whether scATAC-pro includes functions for batch correction.

6) When testing the software, I was stuck at the pre-processing step with the error 'read name does not match' and failed to test the subsequent functions. Besides checking the data (as I noticed the FASTQ file organization seemed to be different between 10x Genomics dataset and scATAC-pro's GitHub description), the author should also provide the example data (e.g. peak matrix) for downstream analysis steps in case the pre-processing step did not work in the users' server.

7) For step-by-step usage, the input, output, and related parameters on each step should be explained for non-specialist users. A east-to-understand tutorial is very important since the integrated workbench should be mostly used by non-specialists who are not familiar with scATAC-seq.

Minor points:

1) Many R/Bioconductor is required by scATAC-pro, but the R configuration is usually significantly varied from one server to another. Thus, an R script to install and test the required packages is welcomed.

2) A few software dependencies is not specified. As for my server environment (CentOS 7), g++, bzip2, python3-devel, ncurses-devel were required.

3) Related to point 2, scATAC-pro specify only one version of Python (e.g. Python 3.6) but its decencies would require another (e.g. Python 2.7). This introduces troubles during the installation. The docker version is a good choice, but this version is poorly documented for now.

Reviewer 2

Were you able to assess all statistics in the manuscript, including the appropriateness of statistical tests used?

Yes, and I have assessed the statistics in my report.

Were you able to directly test the software?

No

Comments to author:

In this manuscript, the authors developed a software named scATAC-pro for analyzing single cell chromatin accessibility sequencing data. As claimed in the manuscript, scATAC-pro can provide multiple options of analysis methods to process different datasets from single cell chromatin accessibility sequencing, including quality control, visualization of the results, in addition to downstream analysis. Overall, this study developed a useful tool for scATAC data processing. However, compared with the other two recent studies that developed similar tools (Nucleic Acids Res. 2019;47:e10; Genome Res. 2019;29:857-69.), this work lacks sufficient novelty and did not address a significant biological question by using the tool.

There are some minor concerns:

- 1) The organization of the manuscript is not clear. Particularly, the section of results about the scATAC-pro "settings" and the corresponding section of methods seem quite redundant. I would suggest that the workflow of scATAC-pro can be moved to the section of "Results", followed by a detailed table, similar to Supplementary Table 1 but listing all options in each step of data analysis. The results should emphasize the default settings and the reasons why these options will be chosen. The methods may have more technical details about each algorithm/software used in the package, for example, what is Louvain algorithm and what is the essential difference between Louvain and other clustering approaches, e.g. K-means?
- 2) As known, whether single cell chromatin accessibility data should be binarized or not is critical in some cases. This also leads to very different choices of normalization methods. It would be better and clearer if scATAC-pro has a specific option like "binary" or not when processing the count matrix followed by two separate paths of "normalization" options to avoid the misuse.
- 3) In the "Dimension reduction", is there any option to choose the number of principal components when using PCA? What is the default setting value?
- 4) Louvain seems to perform little bit better than K-means in most cases. However, it might be helpful to keep the option of K-means as well, especially if the users have a strong sense of the number of cell types they may have.
- 5) More details about the analysis of "differential TF binding between two cell clusters" in the Results should be included, including whether there are options for the cutoffs used to identify differential TFs. Also, it is not clear whether scATAC-pro automatically performs pairwise comparisons between any two clusters or the users can choose the comparison. It must be very useful if scATAC-pro can generate a table for the results including all TFs with significantly differential binding activities in different cell clusters, in addition to the heatmap as shown in Figure 4B which doesn't have statistical evaluations for the comparisons.
- 6) What is the background gene set used for GO analysis here? All genome-wide genes or genes with open chromatin regions only? Is GO analysis for "genes associated with differential

accessibility peaks" (lines 7-9 on page 8) or genes with open chromatin regions in one specific cell cluster, e.g. cluster 0 as shown in Figure 4D?

Reviewer 3

Were you able to assess all statistics in the manuscript, including the appropriateness of statistical tests used?

Yes, and I have assessed the statistics in my report.

Were you able to directly test the software?

Yes

Comments to author:

The authors developed a comprehensive software named scATAC-pro. As they claimed in the manuscript, scATAC-pro can provide multiple options of analysis methods to process different single cell chromatin accessibility sequencing datasets, including quality control, visualization of the results, in addition to downstream analysis. However, I have the following concerns:

1) The organization of the manuscript looks not quite clear, in particular, that the part of results about the scATAC-pro "settings" and the corresponding part of methods seem quite redundant. I would suggest that the workflow of scATAC-pro can be moved to the section of "Results", followed by a detailed table, similar to Supplementary Table 1 but listing all options in each step of data analysis. The results should emphasize the default settings and the reasons why these options will be chosen. The methods may have more technical details about each algorithm/software used in the package, for example, what is Louvain algorithm and what is the essential difference between Louvain and other clustering approaches, e.g. K-means?

2) As we have already known, whether single cell chromatin accessibility data should be binarized or not is critical in some cases. This also leads to very different choices of normalization methods. It would be better and more clear if scATAC-pro has a specific option like "binary" or not when processing the count matrix followed by two separate paths of "normalization" options to avoid the misuse.

3) In the "Dimension reduction", is there any option to choose the number of principal components when using PCA? What is the default setting value?

4) Louvain seems to perform little bit better than K-means in most cases. However, it might be helpful to keep the option of K-means as well, especially if the users have a strong sense of the number of cell types they may have.

5) We expect more details about the analysis of "differential TF binding between two cell clusters" in the Results, including whether there are options for the cutoffs used to identify differential TFs. Also, it is not clear whether scATAC-pro automatically performs pairwise comparisons between any two clusters or the users can choose the comparison. It must be very

useful if scATAC-pro can generate a table for the results including all TFs with significantly differential binding activities in different cell clusters, in addition to the heatmap as shown in Figure 4B which doesn't have statistical evaluations for the comparisons.

6) What is the background gene set used for GO analysis here? All genome-wide genes or genes with open chromatin regions only? Is GO analysis for "genes associated with differential accessibility peaks" (lines 7-9 on page 8) or genes with open chromatin regions in one specific cell cluster, e.g. cluster 0 as shown in Figure 4D?

The software that authors present in the manuscript looks useful for scATAC analysis. It may attract more users if the paper can be re-organized very well including clear explanations of existing options for each analysis step.

Authors Response

Point-by-point responses to the reviewers' comments:

Reviewer #1:

In this manuscript, Yu et al. presented a software package for scATAC-seq. Single cell epigenome sequencing techniques like scATAC-seq is the cutting-edge methods to investigate the functional network beneath the gene expression regulation. However, emerging scATAC-seq platforms and tools may imposed a heavy learning burden on the non-specialists when analyzing scATAC-seq data. Thus, an integrated workbench for scATAC-seq is welcomed. The software package scATAC-pro was designed in comprehensive function, covering nearly the full pipeline of scATAC-seq analysis, from raw read pre-processing to downstream functional/TF analysis. The manuscript is well organized and clearly written. Nevertheless, there are several prominent technical problems to be settled.

Major points:

1) I do not agree with the authors' argument that data binarization of scATAC-seq is not a good choice (as it misses important information). Binarization is helpful to reduce noise and bias in the sparsely mapped scATAC-seq data. The authors would consider to use additional simulated data with noise and/or with more biased mapping to genome regions to evaluate the robustness of their selected methods.

We thank the reviewer for bringing up this issue. We have conducted additional analysis using both simulated data and real data. We first generated simulated data by using a binomial distribution and added different amount of noise. The result is summarized in the figure below. In panel A, we compared the clustering performance of different tools. Here, we can see that some methods work better with non-binarized data (e.g. SCRAT) whereas other methods work better with binarized data (e.g. cisTopic). In panels B and C, we compared the clustering performance of the same clustering algorithm when using both binarized and nonbinarized data. Here we can see most of the time, both Louvain (panel B) and K-mean (panel C) clustering algorithms work better with non-binarized data although at higher noise levels, the performance difference in using the two types of data becomes smaller and in some cases using binarized data is more advantageous. In all cases, we do observe the performance decreases with increasing noise levels.

Supplementary Figure 1. Effect of data binarization on clustering performance using simulated data with different levels of noise. Data was simulated using a binomial distribution of read counts in peaks. Noise level was controlled using the parameter q . $q = 0$ indicates no noise while $q = 1$ indicates the highest level of noise. See methods for details. *cisTopic* and *LSI* can only use binarized data. The rest of the methods use non-binarized data by default although they can also use binarized data. (A) Adjusted rand index (ARI) for different methods, using the FACS-sorted cell types as the ground truth. (B, C) Performance of the same clustering algorithm using binarized and non-binarized data. ARI of Louvain clustering algorithm (B) and *k*-means algorithm (C) using 30 principle components calculated on different numbers of variable peaks.

We also evaluated the performance using a real data set. As can be seen in the figure below, the same conclusion can be reached as with simulated data. That is, for some methods, binarized data gives better performance (e.g. *cisTopic*) whereas for other methods (e.g. *Seurat*) nonbinarized data gives better performance.

Based on this analysis, we have designed our software to enable analysis using both binarized and non-binarized data. We have added the above figure based on simulated data as Supplementary Figure 1 and the above figure based on real data as Supplementary Figure 2D in the revised manuscript. We have also added the following additional text in the Result section.

“For example, using both real and simulated data, we found that the clustering performance using binarized/non-binarized data varies among different methods. For instance, cisTopic performs better using binarized data whereas SCRAT performs better using non-binarized data (Supplementary Figure 1A, Supplementary Figure 2D). Even for the same clustering algorithm, we found that most of the time, both Louvain (Supplementary Figure 1B) and K-mean (Supplementary Figure 1C) clustering algorithms work better with non-binarized data although at higher noise levels, the performance difference becomes smaller and in some cases using binarized data is more accurate. Given this observation, we provide methods that work on either binarized or non-binarized data or both (See details in Methods).”

2) In addition to simulated data, at least one real-world dataset should be used for the method evaluation. Such benchmark settings could be found in a recently published work: Chen et al. Assessment of computational methods for the analysis of single-cell ATAC-seq data. *Genome Biol* 24: 241.

Following the Reviewer’s suggestion, we have evaluated scATAC-pro using the following three real data sets generated with different experimental protocols: 1) a data set of sorted human bone marrow hematopoietic cells generated using the Fluidigm protocol (Buenrostro2018); 2) a data set of human peripheral blood mononuclear cells (PBMCs) generated using the 10x genomics protocol; and 3) a data set of 13 adult mouse tissues generated using the sci-ATAC-Seq protocol (Cusanovich2018). For the sake of brevity, we present the result based on the 10x dataset in the main figures (Figures 2-5). Similar figures based on the two other datasets are presented as supplementary materials (Supplementary Figures 4-9). For benchmarking of clustering performance, we tested different methods on the Buenrostro2018 data since it has the true cell type labels based on FACS sorting which can be used as the ground truth.

3) It is not clear how the authors optimized the parameters or tool selection for different scATAC-seq platforms. I have only noticed the specification of single-end/paired-end reads for this purpose. How about the fitting to different read length / mapping rate / depth etc. between various platforms?

To our knowledge, all current experimental platforms generate short reads, although with different throughput and data quality. Based on the three real datasets (generated with different experimental platforms), we found that the parameter/cutoff selection is crucial for downstream analysis, for example, using different cutoffs to distinguish cell and non-cell barcodes. We recommend users to choose the cutoffs according to the plot of total number of unique reads vs reads in peaks per cell (Figure 3A, Supplementary Figure 6A and 7A). Based on such a plot, we recommend using less stringent cutoff for the fraction of reads in peaks and total unique fragments per cell for data generated using the Fluidigm protocol (Supplementary Figure 6A) and the sci-ATAC-Seq protocol (Supplementary Figure 7A). In contrast, we recommend more stringent cutoffs for 10x data (Figure 3A). scATAC-pro supports specifying different cutoffs in the configuration file.

4) Data visualization functions was not readily accessible to the potential users of scATAC-pro. To this end, automatically calling of genome browser and VisCello tool is preferred, but a step-by-step guideline to configure and run these tools based on scATAC-pro results should be the minimum requirement.

The UCSC and IGV genome browsers are large and complicated programs. In our opinion, it is not necessary to automatically call these genome browsers. The most common approach is to generate genome track files that can be directly used by these genome browsers. This is the functionality we implemented in scATAC-pro.

For using VisCello, we have implemented a new scATAC-pro module called visualize to automatically launch it. We have provided a step-by-step guide in the updated user manual at GitHub repository (<https://github.com/tanlabcode/scATAC-pro>). Briefly, when the parameter prepCello is set to be TRUE in the configuration file, scATAC-pro automatically outputs the input for VisCello when the clustering module is called. By running the visualize module, users can interactively visualize and analyze the data through the VisCello graphical user interface.

We have the following supplementary figure to demonstrate the user interface of VisCello.

Hint: Mouse over points to see the detailed annotation. Drag on plots to select cells. Set plot aesthetics (legend etc.) using cog button on topright.

Hint: Mouse over points to see label.

clusters	number_de_genes
0	2894
1	1785

Showing 1 to 2 of 2 entries

Supplementary Figure 10. Screenshot of the user interface of the visualization tool, VisCello. scATAC-Seq data of human peripheral blood mononuclear cells (PBMCs) was used for illustration purpose. (A) Chromatin accessibility score of peak overlapping with the transcriptional start site of MS4A1 is displayed. Users can use gene name or peak coordinate as the search keyword to explore the accessibility of interested regions. The raw and normalized data can be visualized using uniform manifold approximation and projection (UMAP) or t-distributed

stochastic neighbor embedding (tSNE) with different numbers of principal components. (B) Differential chromatin accessibility analysis. The comparison can be done between any two groups of cells specified by the user. The resulting set of differential accessible regions and the heatmap are downloadable. Shown is the comparison between monocytic cell clusters (clusters 0, 6, 7, 8) versus T cell clusters (clusters 1, 2, 5).

5) Batch effect has been noted in scATAC-seq (e.g. Baker et al., Classifying cells with Scasat, a single-cell ATAC-seq analysis tool. Nucl Acids Res, 2019). It is not clear whether scATAC-pro includes functions for batch correction.

scATAC-pro corrects batch effects using the canonical correlation analysis (CCA, implemented through Seurat v3) in the previous version of the manuscript. In the revised manuscript, we provided two additional methods. The first one is the Harmony algorithm which was shown to group cells based on cell types rather than dataset-specific biases. The second one is designed by us. It directly pools the data and regresses out the confounding effects such as sequence depth per cell and the sample condition. Users can specify the Integrate_By parameter as 'seurat', 'harmony' or 'pool', respectively to choose their batch effect removal method. We have clarified this in the revised manuscript and user manual.

6) When testing the software, I was stuck at the pre-processing step with the error 'read name does not match' and failed to test the subsequent functions. Besides checking the data (as I noticed the FASTQ file organization seemed to be different between 10x Genomics dataset and scATAC-pro's GitHub description), the author should also provide the example data (e.g. peak matrix) for downstream analysis steps in case the pre-processing step did not work in the users' server.

For the case study presented in the main figures, we used the 10x genomics PBMC data in FASTQ format, which was downloaded directly from the 10x genomics website. We did not have trouble using the 10x genomics data set. However, we did run into the same error as the Reviewer did when we ran the Burenstro2018 data. We have figured out that the FASTQ files were corrupted during the downloading process. By re-downloading the data, the preprocess module runs without any error. In addition, we provided a new module named convert10xbam to convert bam files in 10x Genomics format to scATAC-pro format, to facilitate running scATAC-pro using bam files in 10x genomics format.

We have provided the example peak matrix data for all the simulated and real data sets at this link: <https://chopri.app.box.com/s/dlqybg6agug46obiu3mhevofnq4vit4t>.

7) For step-by-step usage, the input, output, and related parameters on each step should be explained for non-specialist users. A east-to-understand tutorial is very important since the integrated workbench should be mostly used by non-specialists who are not familiar with scATAC-seq.

We have provided a more detailed step-by-step tutorial in the user manual on the wiki page of scATAC-pro on GitHub <https://github.com/wbaopaul/scATAC-pro/wiki/Manual>.

Minor points:

1) Many R/Bioconductor is required by scATAC-pro, but the R configuration is usually significantly varied from one server to another. Thus, an R script to install and test the required packages is welcomed.

We thank the Reviewer for the suggestion. We have provided an R script ("install_Rpackages.R") to help install and test the required packages, which can be found on scATAC-pro GitHub website: scripts/install/install_Rpackages.R

2) A few software dependencies is not specified. As for my server environment (CentOS 7), g++, bzip2, python3-devel, ncurses-devel were required.

We thank the Reviewer for identifying those dependencies. We have added those dependencies in the updated user manual.

3) Related to point 2, scATAC-pro specify only one version of Python (e.g. Python 3.6) but its decencies would require another (e.g. Python 2.7). This introduces troubles during the installation. The docker version is a good choice, but this version is poorly documented for now.

Only the TF footprinting analysis module uses Python 2.7. We have provided an installation solution by creating a python 2.7 virtual environment using Miniconda3. During the installation of scATAC-pro, users can skip this installation if they don't need to run TF footprinting analysis module. We have provided more detailed documentation for the docker version in the revision.

Reviewer #2:

In this manuscript, the authors developed a software named scATAC-pro for analyzing single cell chromatin accessibility sequencing data. As claimed in the manuscript, scATAC-pro can provide multiple options of analysis methods to process different datasets from single cell chromatin accessibility sequencing, including quality control, visualization of the results, in addition to downstream analysis. Overall, this study developed a useful tool for ascATAC data processing. However, compared with the other two recent studies that developed similar tools (Nucleic Acids Res. 2019;47:e10; Genome Res. 2019;29:857-69.), this work lacks sufficient novelty and did not address a significant biological question by using the tool.

We agree with the Reviewer that the Scasat (Nucleic Acids Res 2019; 47:e10) and scitools (Genome Res. 2019;29:857-69) have some similar function as scATAC-Pro. However, we would like to point out the following distinct features that the former two software do not have:

Compared to Scasat (Nucleic Acids Res. 2019;47:e10), scATAC-pro has the following innovations:

1) Scasat cannot handle data generated using all existing single cell chromatin accessibility protocols.

Scasat is developed in the Jupyter notebook environment. Although it is interactive, the programming codes are hard to standardize and reuse and users need to customize the analysis step by step.

Scasat binarizes raw peak-by-cell count matrix. As we demonstrate in our analysis in the revised manuscript, the performance depends on specific methods. In contrast, scATAC-pro allow analysis using both binarized and non-binarized data.

Scasat does not provide summary reports data quality assessment.

Most of the analysis tasks in Scasat use one method whereas in scATAC-pro, most of the analysis tasks use more than one methods. This is a unique feature of scATAC-pro that we emphasize in the manuscript. Given that development of analysis algorithms for scATAC-Seq data is still in its early stage, this flexibility in algorithm choice is critical.

Scasat does not provide transcription factor (TF) DNA motif and footprinting analyses. One of the most important utilities of scATAC-Seq data is to understand transcriptional regulation. Without TF motif and footprinting analysis functionalities, Scasat misses a major opportunity of making sense of scATAC-Seq data.

Scasat does not provide analysis for long-range chromatin interactions such as enhancer-promoter interactions.

Scasat does not provide interactive visualization functions for visualizing and exploring ATAC-Seq signals at single-cell and single-gene levels whereas scATAC-pro does so via VisCello. Scasat only provides visualization at cell cluster level.

Compared to scitools (Genome Res. 2019; 29:857-69), scATAC-pro has the following innovations:

scitools is designed for analyzing data generated only by single-cell combinatorial indexing data. It was also mainly developed to facilitate analysis of data generated in the Genome Research publication, instead of a full-fledged software package.

scitools does not provide summary reports for either data quality assessment or downstream analysis.

Most of the analysis tasks in scitools use one method whereas in scATAC-pro, most of the analysis tasks use more than one methods. This is a unique feature of scATAC-pro that we emphasize in the manuscript. Given that development of analysis algorithms for scATAC-Seq data is still in its early stage, this flexibility in algorithm choice is critical.

scitools does not provide transcription factor (TF) DNA motif and footprinting analyses. One of the most important utilities of scATAC-Seq data is to understand transcriptional regulation. Without TF motif and footprinting analysis functionalities, Scitools misses a major opportunity of making sense of scATAC-Seq data.

scitools does not provide Gene Ontology analysis.

scitools does not provide interactive visualization functions.

With regard to the comment on “not address a significant biological question by using the tool”, we respectfully suggest that this is out of the scope of this software manuscript.

Minor concerns:

1) The organization of the manuscript is not clear. Particularly, the section of results about the scATAC-pro "settings" and the corresponding section of methods seem quite redundant. I would suggest that the workflow of scATAC-pro can be moved to the section of "Results", followed by

a detailed table, similar to Supplementary Table 1 but listing all options in each step of data analysis. The results should emphasize the default settings and the reasons why these options will be chosen. The methods may have more technical details about each algorithm/software used in the package, for example, what is Louvain algorithm and what is the essential difference between Louvain and other clustering approaches, e.g. K-means?

We thank the Reviewer for this suggestion. We have restructured the manuscript based on the Reviewer's suggestion. We moved the workflow of scATAC-pro to the Results section. The Results section focuses more on the rationale for the overall design of the software and the choice for the default settings. The Methods section provides more technical details. We also updated the Supplementary Table 1 to list all options for each step. This is also a detailed section for detailed usage of all scATAC-pro modules at the GitHub repository.

2) As known, whether single cell chromatin accessibility data should be binarized or not is critical in some cases. This also leads to very different choices of normalization methods. It would be better and clearer if scATAC-pro has a specific option like "binary" or not when processing the count matrix followed by two separate paths of "normalization" options to avoid the misuse.

We thank the Reviewer for bringing up this issue. As we discussed in our response to Reviewer #1's comment, we have conducted additional simulation studies to evaluate this issue. Our conclusion is that using non-binarized data is better in some cases although in other cases using binarized data does lead to better performance.

Specifically, we generated synthetic data with different levels of noise, and we found some methods using non-binarized data achieve better results (Supplementary Figure 1). We have designed our software to enable analysis using both binarized and non-binarized data. We have added additional text in the Discussion section.

2) In the "Dimension reduction", is there any option to choose the number of principal components when using PCA? What is the default setting value?

In the previous version of the manuscript, we used 30 principle components as the default for PCA. In the revised version, we have added a parameter nREDUCTION in the configuration file to allow users to specify the number of principal components. We have clarified this in the revised manuscript.

3) Louvain seems to perform little bit better than K-means in most cases. However, it might be helpful to keep the option of K-means as well, especially if the users have a strong sense of the number of cell types they may have.

We agree with the Reviewer. We have enabled K-means clustering as an additional option in the revised version.

4) More details about the analysis of "differential TF binding between two cell clusters" in the Results should be included, including whether there are options for the cutoffs used to identify differential TFs. Also, it is not clear whether scATAC-pro automatically performs pairwise comparisons between any two clusters or the users can choose the comparison. It must be very

useful if scATAC-pro can generate a table for the results including all TFs with significantly differential binding activities in different cell clusters, in addition to the heatmap as shown in Figure 4B which doesn't have statistical evaluations for the comparisons.

In the previous version of the manuscript, scATAC-pro supports comparison between any two cell clusters defined by the user and uses a p-value cutoff of 0.05 to identify differentially bound TFs. In the revised manuscript, we enable both one cluster vs one cluster comparison and one cluster vs rest of clusters comparison. We also provide a parameter (pvalue_fp) in the configuration file to specify p-value cutoff for differentially bound TFs. We provide the results showing differential bound TFs for each cluster both as a table and as a heatmap (e.g. Figure 4C) as suggested by the Reviewer.

5) What is the background gene set used for GO analysis here? All genome-wide genes or genes with open chromatin regions only? Is GO analysis for "genes associated with differential accessibility peaks" (lines 7-9 on page 8) or genes with open chromatin regions in one specific cell cluster, e.g. cluster 0 as shown in Figure 4D?

The differential chromatin accessibility analysis can be conducted between two groups of cell clusters; a group can contain one or more cell clusters, which can be specified in the configuration file. Therefore, the GO analysis is for genes associated with differential accessibility peaks in one cell group. The background gene set is comprised of all genes associated with the differential accessibility peaks in all cell groups generated by the differential accessibility analysis.

Reviewer #3:

The authors developed a comprehensive software named scATAC-pro. As they claimed in the manuscript, scATAC-pro can provide multiple options of analysis methods to process different single cell chromatin accessibility sequencing datasets, including quality control, visualization of the results, in addition to downstream analysis. However, I have the following concerns:

1) The software that authors present in the manuscript looks useful for scATAC analysis. It may attract more users if the paper can be re-organized very well including clear explanations of existing options for each analysis step.

We have re-organized the manuscript as suggested by Reviewers 2 and 3.

2) The organization of the manuscript looks not quite clear, in particular, that the part of results about the scATAC-pro "settings" and the corresponding part of methods seem quite redundant. I would suggest that the workflow of scATAC-pro can be moved to the section of "Results", followed by a detailed table, similar to Supplementary Table 1 but listing all options in each step of data analysis. The results should emphasize the default settings and the reasons why these options will be chosen. The methods may have more technical details about each algorithm/software used in the package, for example, what is Louvain algorithm and what is the essential difference between Louvain and other clustering approaches, e.g. K-means?

This is the same comment as by Reviewer 2. Please see our response to Reviewer 2's minor comment #1.

3)As we have already known, whether single cell chromatin accessibility data should be binarized or not is critical in some cases. This also leads to very different choices of normalization methods. It would be better and more clear if scATAC-pro has a specific option like "binary" or not when processing the count matrix followed by two separate paths of "normalization" options to avoid the misuse.

This is the same comment as by Reviewer 1. Please see our response to Reviewer 1's major comment #1.

4)In the "Dimension reduction", is there any option to choose the number of principal components when using PCA? What is the default setting value?

This is the same comment as by Reviewer 2. Please see our response to Reviewer 2's minor comment #2.

5)Louvain seems to perform little bit better than K-means in most cases. However, it might be helpful to keep the option of K-means as well, especially if the users have a strong sense of the number of cell types they may have.

This is the same comment as by Reviewer 2. Please see our response to Reviewer 2's minor comment #3.

6)We expect more details about the analysis of "differential TF binding between two cell clusters" in the Results, including whether there are options for the cutoffs used to identify differential TFs. Also, it is not clear whether scATAC-pro automatically performs pairwise comparisons between any two clusters or the users can choose the comparison. It must be very useful if scATAC-pro can generate a table for the results including all TFs with significantly differential binding activities in different cell clusters, in addition to the heatmap as shown in Figure 4B which doesn't have statistical evaluations for the comparisons.

This is the same comment as by Reviewer 2. Please see our response to Reviewer 2's minor comment #4.

7) What is the background gene set used for GO analysis here? All genome-wide genes or genes with open chromatin regions only? Is GO analysis for "genes associated with differential accessibility peaks" (lines 7-9 on page 8) or genes with open chromatin regions in one specific cell cluster, e.g. cluster 0 as shown in Figure 4D?

This is the same comment as by Reviewer 2. Please see our response to Reviewer 2's minor comment #5.

Second round of review

Reviewer 1

In the revised manuscript, the selection of algorithm and parameters in scATAC-pro have been supported by additional evidence. Besides, the software can be much more smoothly ran and the documentation has been significantly improved. Overall, the authors have addressed all my previous concerns.

Reviewer 3

The revised version has been significantly improved. The authors have responded to all my previous review comments/questions. Again, since the scATAC-seq technology is relative new, almost none of existing software can cover complete analyses for the data generated by different experimental protocols and none of them have the functions to satisfy users by providing flexible options at each step. Now the software scATAC-pro described in this paper fills the gap with multiple necessary options for the data analyses, from the most important step at very beginning, data quality control, to integrative downstream biological analyses and data visualization. Particularly the scATAC-pro comes up with comprehensive quality assessment of the scATAC-seq data generated by ALL existing experimental protocols, in addition to carefully evaluated analysis methods for all major aspects which the scATAC-seq explored. I believe that this software/tool will be widely used by the community especially considering the rapidly increasing volume and complexity of scATAC-seq data within coming years.

A few minor concerns/typos are listed below.

(1) How to connect TF motif enrichment analysis with footprint analysis? Of course, it is not necessary to make a conclusive statement between them because of different experimental designs and study purposes. But it might be helpful if the authors can explain a little more in the workflow or in the case study to avoid over-interpretation of the results in case users have no clear concepts about these analyses.

(2) How many cell types did the authors identify in the case study, e.g. Figure 4? There seems 10 groups totally (from group 0 to 9). Please correct the number in lines 27-28 (pg 12), "we found 9 cell types", and add the group c9 in Figure 4C.

(3) Would it become better if Figure 4E will be moved to under Figure 4D so that more spaces can be left for Figures 4B and C? It is hard to see the TF names clearly in Figure 4C now.

Authors Response

Point-by-point responses to the reviewers' comments:

(1) How to connect TF motif enrichment analysis with footprint analysis? Of course, it is not necessary to make a conclusive statement between them because of different experimental designs and study purposes. But it might be helpful if the authors can explain a little more in the workflow or in the case study to avoid over-interpretation of the results in case users have no clear concepts about these analyses.

We provided the following description for the difference between TF motif enrichment analysis and footprinting analysis in the "Transcription factor footprinting analysis" subsection in the Results section.

"ATAC-seq and related technologies use the Tn5 enzyme to cleavage nucleosome-free DNA while keeping the transcription factor binding sites intact due to protection by the bound TF. As a result, a small region, referred to as the footprint, exhibits reduced Tn5 cleavage rate at the ATAC-Seq peak locus. Unlike TF motif enrichment analysis, TF footprinting analysis provides direct evidence of TF binding to the chromatin".

(2) How many cell types did the authors identify in the case study, e.g. Figure 4? There seems 10 groups totally (from group 0 to 9). Please correct the number in lines 27-28 (pg 12), "we found 9 cell types", and add the group c9 in Figure 4C.

We identified 10 cell types. We have corrected the typo on page 12.

We didn't find any enriched TF footprinting for group c9. Therefore group c9 is not shown in Figure 4C.

(3) Would it become better if Figure 4E will be moved to under Figure 4D so that more spaces can be left for Figures 4B and C? It is hard to see the TF names clearly in Figure 4C now.

Thank you for the suggestion. We have modified the figure as suggested.